# Radiochemical Synthesis of 4-[^18^F]FluorobenzylAzide and Its Conjugation with EGFR-Specific Aptamers

**DOI:** 10.3390/molecules28010294

**Published:** 2022-12-30

**Authors:** Viktor A. Il’in, Elena V. Pyzhik, Anton B. Balakhonov, Maksim A. Kiryushin, Evgeniya V. Shcherbatova, Andrey A. Kuznetsov, Pavel A. Kostin, Andrey V. Golovin, Vladimir A. Korshun, Vladimir A. Brylev, Kseniya A. Sapozhnikova, Alexey M. Kopylov, Galina V. Pavlova, Igor N. Pronin

**Affiliations:** 1Burdenko National Medical Research Center of Neurosurgery, Ministry of Health of the Russian Federation, 125047 Moscow, Russia; 2Chemistry Department, Lomonosov Moscow State University, 119991 Moscow, Russia; 3Shemyakin-Ovchinnikov Institute of Bioorganic Chemistry, Russian Academy of Sciences, 117997 Moscow, Russia; 4Medical Genetics Department, Sechenov First Moscow State Medical University, 119991 Moscow, Russia; 5Laboratory of Neurogenetics and Genetics Development, Institute of Higher Nervous Activity and Neurophysiology of Russian Academy of Sciences, 117485 Moscow, Russia

**Keywords:** CuAAC, oligonucleotides, PET imaging, EGFR, iodonium ylides

## Abstract

Central nervous system tumors related to gliomas are of neuroectodermal origin and cover about 30% of all primary brain tumors. Glioma is not susceptible to any therapy and surgical attack remains one of the main approaches to its treatment. Preoperative tumor imaging methods, such as positron emission tomography (PET), are currently used to distinguish malignant tissue to increase the accuracy of glioma removal. However, PET is lacking a specific visualization of cells possessing certain molecular markers. Here, we report an application of aptamers to enhancing specificity in imaging tumor cells bearing the epidermal growth factor receptor (EGFR). Glioblastoma is characterized by increased EGFR expression, as well as mutations of this receptor associated with active division, migration, and adhesion of tumor cells. Since 2021, EGFR has been included into the WHO classification of gliomas as a molecular genetic marker. To obtain conjugates of aptamers GR20 and GOL1-specific to EGFR, a 4-[^18^F]fluorobenzylazide radiotracer was used as a synthon. For the production of the synthon, a method of automatic synthesis on an Eckert & Ziegler research module was adapted and modified using spirocyclic iodonium ylide as a precursor. Conjugation of 4-[^18^F]fluorobenzylazide and alkyne-modified aptamers was carried out using Cu(I)-catalyzed azide-alkyne cycloaddition (CuAAC) with/without the TBTA ligand. As a result, it was possible to obtain ^18^F-labelled conjugates with 97% radiochemical purity for [^18^F]FB-GR20 and 98% for [^18^F]FB-GOL1. The obtained conjugates can be used for further studies in PET analysis on model animals with grafted glioblastoma.

## 1. Introduction

Gliomas are the most common primary brain tumors, characterized by pronounced proliferation and aggressive infiltration [1].

The standard treatment tactic is surgical removal of the maximum volume of the tumor, after which radiation therapy and/or chemotherapy are performed [2]. The estimated 5-year survival rate among patients with glioma was 34.9%, whereas for glioblastoma the prognosis was only 5.5% [3]. Glial tumors are characterized by significant heterogeneity, infiltration growth and lack of clear boundaries, which makes glioma neuroimaging challenging [4]. More accurate diagnostic methods are needed in order to perform effective treatment and increase patient survival. Regarding PET, accuracy improvement means specific visualization of cells with certain molecular markers.

The epidermal growth factor receptor (EGFR) is a known biomarker of gliomas of high malignancy, and, since 2021, it has been included into the WHO classification of gliomas [5]. This transmembrane protein of the tyrosine kinase family is highly expressed in glioblastoma, compared to low-grade tumors. Such overexpression further affects cell migration and increases tumor aggressiveness. EGFR is a significant cancer marker for diagnosis and a promising target for the treatment of glioblastoma, a malignant brain tumor.

Currently, diagnostic and therapeutic tools based on aptamers are of particular interest. Aptamers are unique single-stranded nucleic acid molecules. They exhibit high affinity and specificity to certain target compounds (e.g., proteins, peptides) which they are able to bind with. For this reason, they are also called “chemical antibodies”. A design of an aptamer specific to the epidermal growth factor receptor is in great demand, among other possible applications of aptamers.

These molecules also act as promising compounds for use in nuclear medicine [6]. Technologies based on aptamers are employed in the diagnostic methods of single-photon emission computer tomography (SPECT) and PET. The key step in using aptamers for SPECT and PET is conjugation, i.e., binding the aptamers to a radioactively labelled tracer, with isotope fluorine-18 being the most preferable one.

To date, a number of aptamers with radioactive labels are reported to be in use for the visualization of malignant tumors [6]. Radiochemistry methods for the visualization of aptamers have become widely applicable, and the importance and necessity of this kind of research has been noted [7].

A number of methods for obtaining [^18^F]-labelled ligands to be conjugated with aptamers for their use in PET diagnostics are described to date [8,9,10,11,12]. These techniques utilized various click chemistry approaches for obtaining fluorine-18-labelled aptamers. For example, 4-[^18^F]fluorobenzylazide [10] can be used in conjugation with aptamers or proteins. In this work, a spirocyclic compound of hypervalent iodine (III) was used as a precursor [13]. Iodine (III) derivatives of spirocyclic malonates were introduced in 2014 by a research group at the Department of Radiochemistry and Biomarkers at Harvard Medical School as convenient precursors of ^18^F in aromatic compounds [14].

Our research is aimed at the development of radioligands conjugated with EGFR-specific single-stranded DNA aptamers for use in PET imaging for diagnostics of brain tumors which overexpress EGFR. Since 4-[^18^F]fluorobenzylazide is rapidly synthesized and gives a high yield when conjugated with various aptamers, it was chosen as the radioactive label for obtaining [^18^F]-labelled aptamers.

Here we present a study of aptamer conjugation using a copper(I)-catalyzed azide-alkyne cycloaddition (CuAAC) technique with/without the TBTA ligand to label aptamers specific to EGFR [15].

## 2. Results

Automated radiochemical synthesis of 4-[^18^F]fluorobenzylazide was performed using an Eckert & Ziegler research module based on the scheme depicted below in Materials and methods. The spirocyclic iodoniumylide precursor was used in a direct 1-step radiofluorination to afford radiotracer, followed by copper-mediated click-chemistry conjugation to an aptamer.

After 12 repeats, we achieved a persistent preparation of synthon with radiochemical yield in the range of 25.32 ± 3.36% (n.d.c.). The activity of 4-[^18^F]fluorobenzylazide was 7.4–17.1 GBq based on initial [^18^F]F-activity (29.4–70.8 GBq). The molar activity of the tracer was in the 246.1–359.3 GBq/μmol range (Mw = 151.1 g/mol). Radiochemical purity was determined by HPLC to be in the range from 96.4 to 98.9% (Figure 1). Having achieved stable results for the tracer synthesis, we started to perform the conjugation with aptamers.

The production of the radioactive synthon and its conjugation with aptamers are shown in Figure 2. Five conjugations of 4-[^18^F]fluorobenzylazide (177–381 MBq) with aptamer GR20 were performed using CuAAC without the TBTA ligand. As a result, the [^18^F]FB-GR20 conjugate (112–212 MBq) was obtained, with radiochemical purity in the 95.6–98.32% range (Figure 3A,C).

The resulting radioactive conjugate was purified on a NAP-5 column to remove the residual components of the reaction mixture and extract the labelled aptamer. Acceptable residue solvent content in the conjugate and maximum activity for the combined second and third fractions were obtained in the process of purification of aptamer GR20 conjugate. Yield of the labelled conjugate after purification on NAP-5 ranged from 57 to 72%, (n.d.c.) based on 4-[^18^F]fluorobenzylazide.

CuAAC conjugation without the TBTA ligand was also tested for the GOL1 aptamer (105–128 MBq). However, HPLC conjugate analysis showed an unsatisfactory result (Figure 3B,D). The GR20 conjugate was, to a lesser extent, also degraded, as can be seen on the chromatograms (Figure 3C).

To use radioactive conjugates in PET diagnostics, it is necessary to minimize their destruction. In order to increase the stability of the conjugates, we decided to try another click chemistry conjugation method, which utilizes a complex of copper with the TBTA ligand, and additional coprecipitation with acetone in the presence of lithium perchlorate instead of purification on the NAP-5 column.

The GOL1 conjugate (138–205 MBq) obtained using this approach appeared to be quite stable (*n* = 3). Stability was monitored after 2 h at room temperature via HPLC analysis. Radiochemical purity was more than 98% (Figure 4A). Yield of the labelled conjugate after coprecipitation was in the range from 68 to 74%, based on non-decay corrected 4-[^18^F]fluorobenzylazide (204–302 MBq). Molar activity of the labelled aptamer, calculated by dividing the observed activity by the molar concentration of the GOL1 conjugate (Mw = 15,496.1 g/mol) used for the reaction, was 8.9–13.6 GBq/μmol.

The same technique applied to GR20 conjugation (106–177 MBq) also resulted in a rather stable conjugate (*n* = 3). Radiochemical purity was more than 97% (Figure 4B). Radiochemical yield of the [^18^F]FB-GR20 conjugate after coprecipitation ranged from 61 to 67%, (n.d.c.) based on 4-[^18^F]fluorobenzylazide (159–290 MBq). Molar activity of the labelled aptamer, calculated by dividing observed activity by the molar concentration of the GR20 conjugate (Mw = 14,543.1 g/mol) used for the reaction, was 6.5–11.8 GBq/μmol.

## 3. Discussion

Automatic synthesis of 4-[^18^F]fluorobenzylazide has been performed and the stability of the conjugated aptamers has been studied. The product radiochemical yield of the product was similar to that reported previously [14].

Using the conjugation technique with the TBTA ligand, we managed to conjugate the product with aptamers GR20 and GOL1 specific to EGFR (Figure 2). During the study, we compared the stability of the obtained [^18^F]FB-GR20 and [^18^F]FB-GOL1conjugates.

Secondary structure of aptamer GR20 is shown in Figure 5. This aptamer, described in the work of Zavyalova et al. [15], was obtained by truncating the aptamer U31, which was previously selected as showing better affinity for the EGFR and its mutant isoform EGFR_vIII_ [16]. The latter is the most common in glioblastoma and correlates with a poor prognosis for patients [16]. Unfortunately, we failed to obtain a GR20 conjugate of appropriate radiochemical purity and stability using CuAAC without the TBTA ligand and purification on the NAP-5 column (Figure 3C).

The damaging effect of Cu^2+^ ions on DNA, and the secondary structure of aptamers rich in CpG bonds, is well known [17,18]. Using the conjugation technique without the TBTA ligand, we speculate that incomplete removal of Cu(I) can occur after addition of ascorbate solution, and copper can likely undergo oxidation by atmospheric oxygen. Hence, copper (II) interacting with the aforementioned fragments stabilizes a certain spatial structure of the oligonucleotide (quadruplex), which, in turn, hinders the reactivity of the alkyne group due to steric effects. For the labeled GR20 conjugate, this effect likely manifested by the presence of a considerable “shoulder” next to the peak on the chromatogram (Figure 3A,C).

We suggest that the same effect explains the impossibility of obtaining a labeled conjugate of aptamer GOL1 by the CuAAC method which utilizes copper sulfate. As seen from the structure of GOL1 (Figure 6), there are significantly more CpG sites in it than in GR20. Hence, the molecule of aptamer GOL1 is more prone to being influenced byCu^2+^ ions during conjugation. As a result, we speculate that it adopts the shape of a quadruplex, which is less suitable and undesirable for azide-alkyne coupling.

On the other hand, the conjugation technique with the TBTA ligand can more likely eliminate the generation of copper(II). Presumably, complex Cu-TBTA inhibit the oxidation of Cu(I) due to more complete copper removal after coprecipitation with lithium perchlorate. Considering the method of conjugation utilizing TBTA, the negative effect of copper on CpG-rich oligonucleotide sites is eliminated by the presence of the complex with the TBTA ligand. This method was developed in 2004 by a scientific group from the Scripps Research Institute in California [19], which studied the possibility of stabilizing copper (I) during catalysis in order to prevent its oxidation and disproportionation, as well as to reduce its negative effect on oligonucleotides during conjugation through azide-alkyne cycloaddition. The researchers used various polytriazoles in their work, and TBTA (tris-(benzyltriazolylmethyl)amine) proved to be the most effective one.

After testing the conjugation technique using CuSO_4_ and not achieving the desired results, we decided to test a Cu-TBTA premix. The first trial conjugations were performed with the same proportions of reagents as in the method with copper sulfate. However, a suspension was formed in the reaction mixture when it was heated to 37 °C. To eliminate this problem, DMSO was added for better solubility of the aptamers and the concentration of sodium ascorbate was proportionally changed.

The use of the NAP-5 column for purification in the method with TBTA did not allow the conjugate to be isolated, apparently, it remained bound to the column. Alternatively, instead of purification, we co-precipitated the conjugate with acetone in the presence of lithium perchlorate. Since the premix was dissolved in dimethyl sulfoxide, an additional modification was also made in the radiochemical synthesis procedure, namely, we replaced acetonitrile with DMSO.

It should be noted, that during the experimental work (specifically, the analysis of labelled conjugates), we moved away from the HPLC method described in literature which utilizes a C8 column, replacing it with a C18 column and using a gradient with a higher concentration of acetonitrile, which allowed us to improve the resolution of peaks on the chromatogram and reduce the analysis time.

The use of aptamers in the diagnosis and therapy of oncological diseases and the study of their properties are currently developing intensively [20]. In our work, the production of radioactive conjugates with aptamers implies their subsequent use in experiments on model animals. This requires the labelled aptamer to be stable for a certain period of time and not to degrade. For the ME07 aptamer conjugate described previously [20], a rapid decrease in tumor accumulation was shown in the 30–90 min time interval after injection, which authors partially attribute to the degradation and disintegration of the conjugate. The authors of the work recommend PET scanning at the early stage (up to 30 min) after injection into animals. With this in mind, we investigated how the conjugated aptamers behave 2 h after synthesis. A slight decrease in radiochemical purity to 96.8% was noted for aptamer GOL1 prepared by the Cu-TBTA method (Figure 4C). Similar results were obtained for the GR20 aptamer: the radiochemical purity of the conjugate after 2 h was 94.8% (Figure 4B). In the case of using the CuAAC technique without TBTA, the obtained purified conjugates showed signs of decay, as indicated by the presence of “shoulders” and the shapes of the peaks on the chromatograms (Figure 3C,D). Analyzing the data for radiochemical purity and stability for both cases, we can conclude that the use of the TBTA ligand and the subsequent coprecipitation with acetone allows us to achieve more acceptable results. The high stability ofthe aptamer conjugates is an important factor for further in vivo studies.

## 4. Materials and Methods

### 4.1. General

Fluorine-18 was obtained usingan Eclipse RDS cyclotron (Siemens) by irradiation of oxygen-18 with protons with 11 MeVenergy. Radiochemical synthesis and purification of 4-[^18^F]fluorobenzylazide was carried out on an EC-290 research module (Eckert & Ziegler). HPLC analyses were performed on an Agilent 1290 Infinity II chromatograph with a diode-matrix detector and a Gabi Star radioactivity detector and an Agilent 1260 Infinity I chromatograph with a variable wavelength detector and a Flow-RAM radioactivity detector.

### 4.2. Aptamers

Aptamer GR20 [13], consisting of a sequence of 46 nucleotides (ACG CAC CAT TTG TTT AAT ATG TTT TTT AAT TCC CCT TGT GGT GTG T), and aptamer GOL1, consisting of a sequence of 49 nucleotides (GCC GGC ATT TTG ACG CCG CCC CGG CTG CTT ATG CTC CGG GGC ATA TGG C), (patent No. 2022133936) were synthesized by GENTERRA as 5′-alkyne derivatives.

Both aptamers have an increased affinity to a functionally significant molecular target of protein nature (the extracellular domain of the EGFR protein), as shown in [15].

### 4.3. Quality Control

#### 4.3.1. Quality Control 4-[^18^F]FluorobenzylAzide

Quality control of the obtained 4-[^18^F]fluorobenzylazide was carried out by HPLC on an Agilent 1290 Infinity II chromatograph with a diode-matrix detector and a Gabi Star radioactivity detector on a Phenomenex Luna18 100 Å (5 µm, 4.6 × 250 mm) column using a mobile phase of 55% acetonitrile and 45% 0.1 M ammonium formate at the flow rate of 1 mL/min. Product retention time was 11.5 ± 0.2 min. Analysis was carried out with an ultraviolet detector at the 254 nm wavelength.

#### 4.3.2. Quality Control of Conjugates and Aptamers

Quality control was carried out by HPLC on an Agilent 1260 chromatograph onan Phenomenex Luna C8 column (5 µm, 4.6 × 150 mm) using a gradient starting with 90% eluent A (0.1 M triethylammonium acetate in water) and 10% solvent B (acetonitrile) with replacement by 70% eluent A and 30% eluent B at 35 min at the flow rate of 1 mL/min. The wavelength of the ultraviolet detector was 254 nm.

Or on a Phenomenex Luna C18 column (5 µm, 4.6 × 250 mm), using a gradient starting with 100% eluent A (0.1 M triethylammonium acetate in water) and 2% solvent B (acetonitrile) with replacement by 65% eluent A and 45% eluent B at 20 min at the flow rate of 1 mL/min. The wavelength of the ultraviolet detector was 254 nm.

### 4.4. Radiochemical Synthesis

We adapted the synthesis technique described in [8] to obtain the radioactive synthon 4-[^18^F]fluorobenzylazide using an Eckert & Ziegler research module, according to the following scheme (see Figure 7).

A precursor based on a spirocyclic hypervalent iodine (III) derivative was used for the synthesis of the radioactive tracer 4-[^18^F]fluorobenzylazide. It was obtained according to the method also described in [8].

### 4.5. Conjugation of 4-[^18^F]Fluorobenzyl Azide with Aptamers

The solution of 4-[^18^F]fluorobenzylazide in a solvent (acetonitrile, DMSO) was used immediately after synthesis and quality analysis. The reagents involved in the process were mixed sequentially with essential mixing on avortex after adding each reagent. Two conjugation protocols were used in the current work.

#### 4.5.1. Conjugation of Aptamers without Cu-TBTA

The reagents used in this technique are a 140 mM water solution of copper (II) sulfate, a 140 mM solution of sodium ascorbate in a 0.1 M borate buffer, a and 0.5 mM water solution of aptamer. The reagents were manually mixed in an Eppendorf tube as follows: 20 μL of aptamer solution, 20 μL of copper (II) sulfate, 200 μL of sodium ascorbate, 50 μL of tracer solution in acetonitrile and stirred on a vortex. After mixing, the reaction mixture was incubated with slow stirring (250 rpm) under heating (37 °C) in aPS100C thermostatically controlled desktopshaker for 15 min. Next, the conjugate was purified by solid-phase extraction on a NAP-5 column.The second and third fractions corresponding to the labelled aptamer and containing the least number of impurities were collected and combined.

#### 4.5.2. Conjugation of Aptamers with Cu-TBTA

The reagents used in this technique are a 50 mM solution of the copper-TBTA premix, a 280 mM solution of sodium ascorbate in 0.1 M of triethylammonium acetate, a DMSO and a 0.5 mM water solution of aptamer. The reagents were manually mixed in an Eppendorf tube as follows: 20 μL of aptamer solution, 40 μL of copper-TBTA premix, 75 μL of sodium ascorbate, 50 μL of DMSO, 50 μL of tracer solution in DMSO, and stirred on a vortex. After mixing, the reaction mixture was incubated with slow stirring (250 rpm) under heating (37 °C) in aPS100C thermostatically controlled desktop shaker for 15 min. Further purification of the conjugate was not required.

#### 4.5.3. Coprecipitation of Conjugates with Acetone in the Presence of Lithium Perchlorate (LiClO_4_)

50 µL of lithium perchlorate were added to the resulting conjugate and mixed on a vortex; 1.3 mL of acetone were added to the reaction mixture. The tube was stirred and centrifuged for 5 min at 10,000 rpm. The top layer of acetone was discarded. The remaining reaction mixture was incubated in a thermoshaker for 5 min at 40 °C to eliminate acetone residues.

Radiochemical purity of the obtained conjugates labeled with fluorine-18 was determined by HPLC.

## 5. Conclusions

The aptamers GR20 and GOL1 have high affinity for the EGFR receptor, whose enhanced expression is characteristic for human glioblastoma [21,22]. These aptamers can increase the specificity of the diagnosis using PET. In our work, we have tested various methods of azide-alkyne cycloaddition with/without the use of TBTA ligand. We have obtained radioactive conjugates of aptamers GR20 and GOL1, and showed their stability persisting for 2 h using TBTA as an additive in conjugation experiments. The reproducibility of the experimental results and the stability of the conjugates obtained in the work will allow carrying out research involving laboratory animals. In future work, we plan to study the accumulation of aptamers in highly malignant glioma cells and to obtain more detailed information about their interaction with molecular genetic markers.

## 6. Patents

Aptamer GOL1 used in our research is patented in the Federal Institute of Industrial Property (FIPS), No. 2022133936: “Aptameric deoxyribooligonucleotide specifically binding to EGFR and EGFR vIII” (A.V. Golovin, A.M. Kopylov, G.V. Pavlova, L.V. Fab, A.I. Alekseeva, D.Y. Usachev, I.N. Pronin).

## Figures and Tables

**Figure 1 molecules-28-00294-f001:**
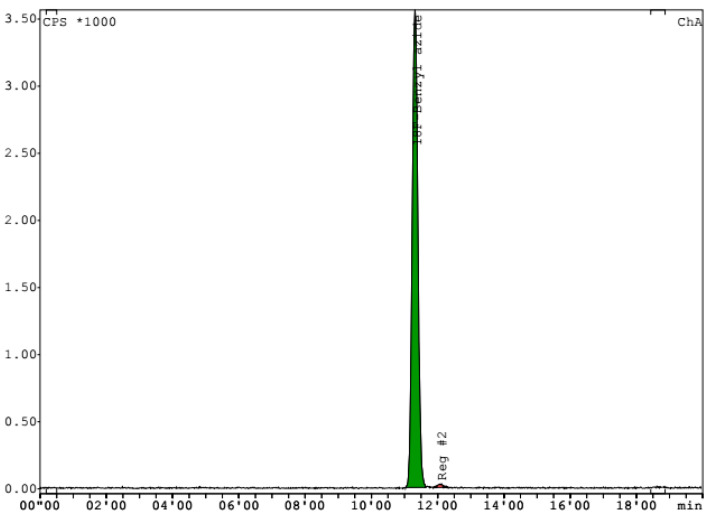
Chromatogram of 4-[^18^F]fluorobenzylazide after synthesis. Phenomenex Luna18 100 Å (5 µm, 4.6 × 250 mm) column; mobile phase: 55% acetonitrile and 45% 0.1 M ammonium formate; flow rate 1 mL/min.

**Figure 2 molecules-28-00294-f002:**
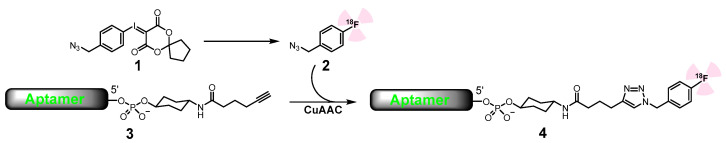
Preparation of the aptamer conjugates using 4-[^18^F]fluorobenzylazide by the CuAAC method with/without TBTA (**1**-spyrocyclic iodonium ylide precursor, **2**-4-[^18^F]fluorobenzylazide, **3**-5′-alkyne-modified aptamer, **4**-radioactive conjugate).

**Figure 3 molecules-28-00294-f003:**
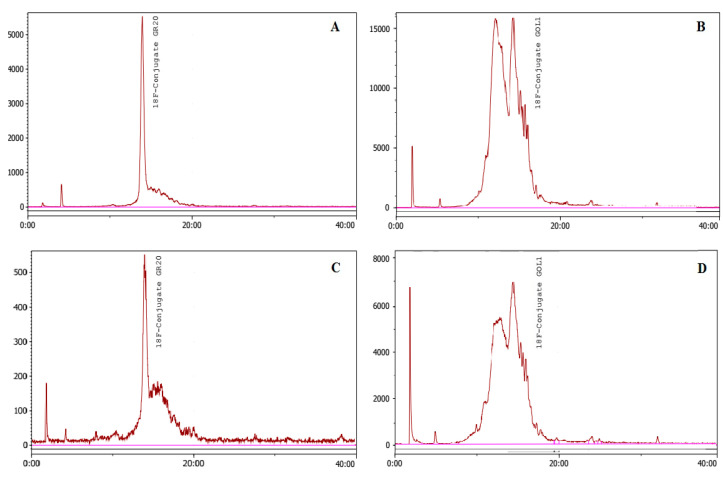
Click reaction without the TBTA ligand. HPLC profiles of the [^18^F]FB-GR20 conjugate ((**A**)—before and (**C**)—after purification on NAP-5) and the [^18^F]FB-GOL1 conjugate ((**B**)—before and (**D**)—after purification on NAP-5). Conditions: Phenomenex Luna C8 column (5 µm, 4.6 × 150 mm); mobile phase: linear gradient starting with 90% eluent A (0.1 M triethylammonium acetate in water) and 10% solvent B (acetonitrile) with replacement by 70% eluent A and 30% eluent B at 35 min; flow rate 1 mL/min.

**Figure 4 molecules-28-00294-f004:**
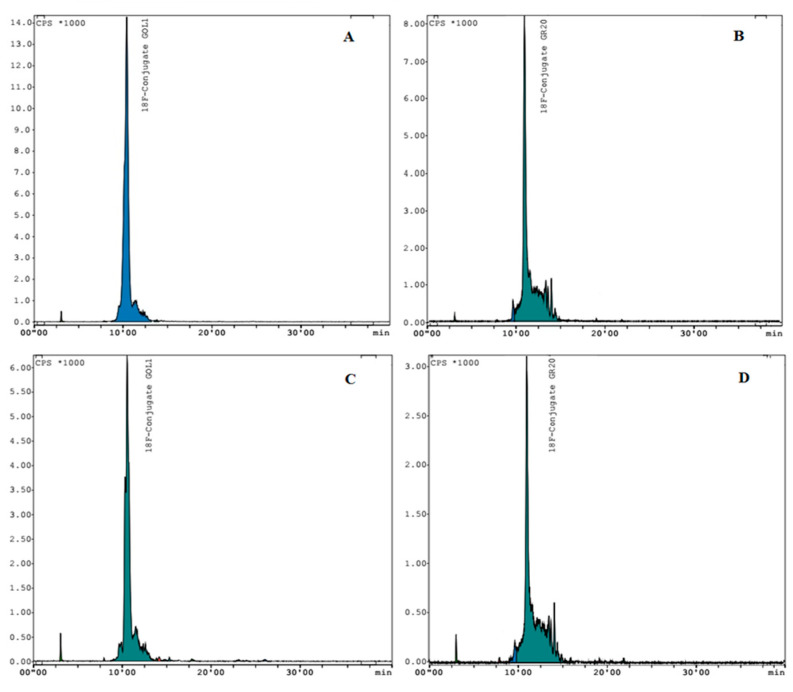
Click reaction with the TBTA ligand. HPLC profiles of precipitated GOL1 conjugate ((**A**)—immediately after conjugation and (**C**)—after 2 h) and precipitated GR20 conjugate ((**B**)—immediately after conjugation and (**D**)—after 2 h). Conditions: Phenomenex Luna C18 column (5 µm, 4.6 × 250 mm); mobile phase: linear gradient: starting with 100% eluent A (0.1 M triethylammonium acetate in water) and 2% solvent B (acetonitrile) with replacement by 65% eluent A and 45% eluent B at 20 min, flow rate 1 mL/min.

**Figure 5 molecules-28-00294-f005:**
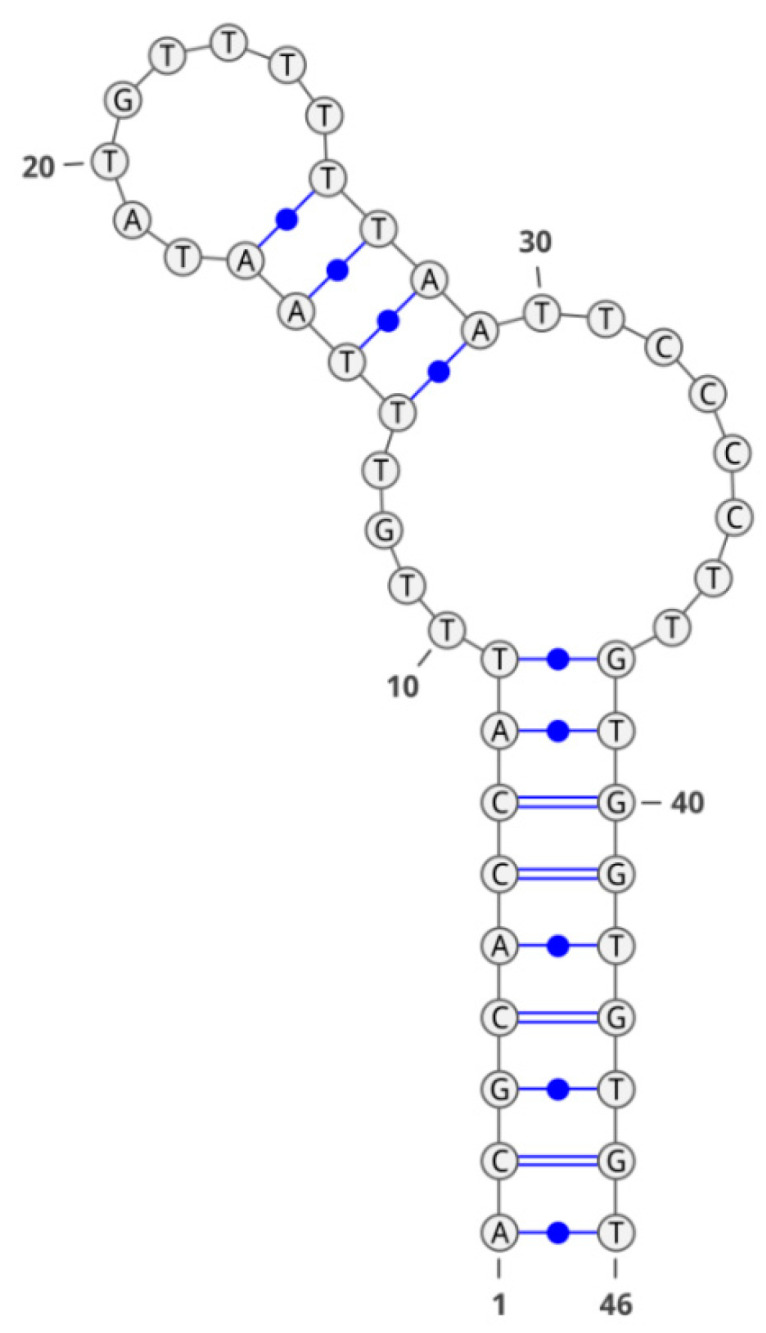
Secondary structure of the GR20 aptamer.

**Figure 6 molecules-28-00294-f006:**
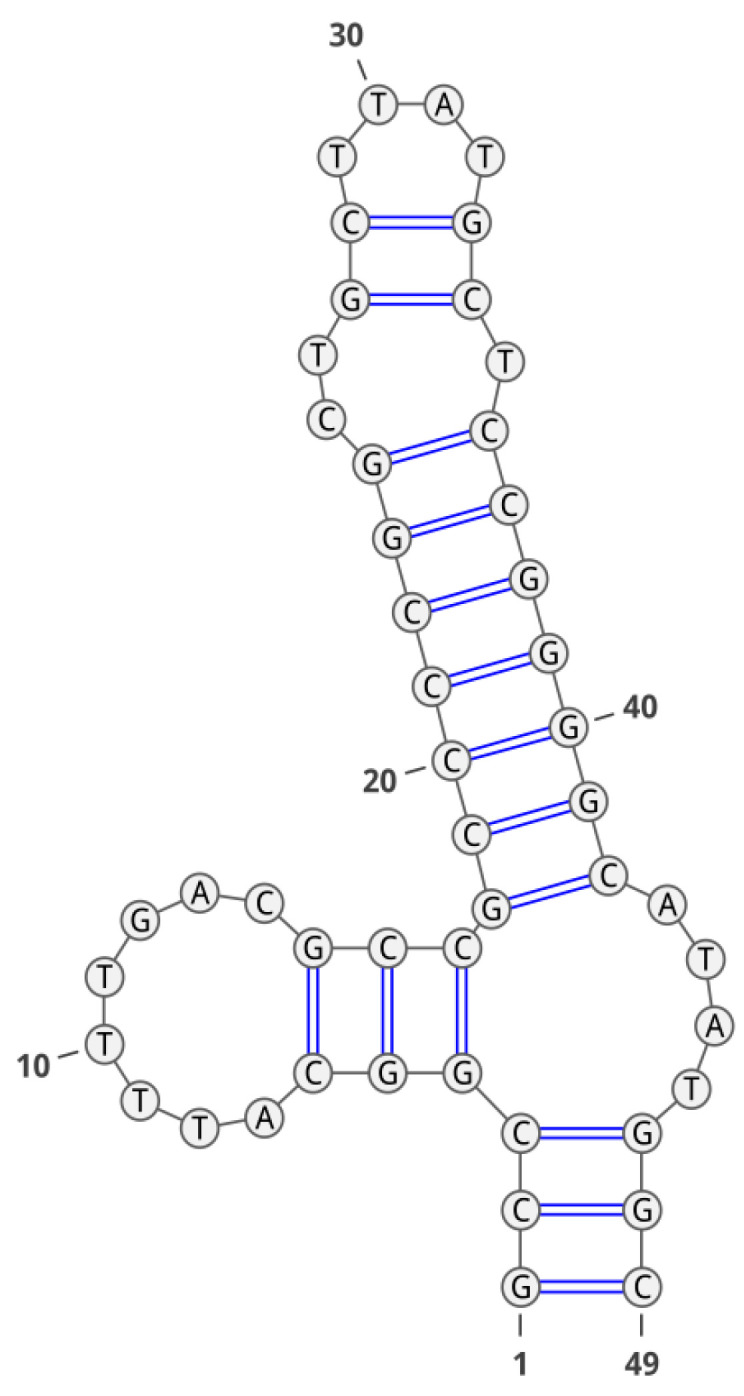
Secondary structure of the GOL1 aptamer.

**Figure 7 molecules-28-00294-f007:**
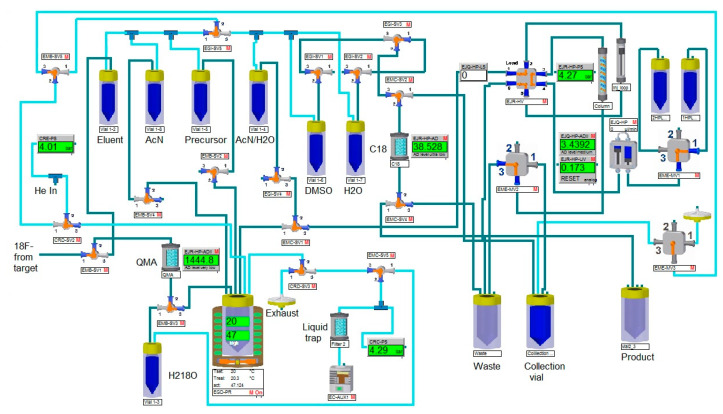
The synthesis scheme of 4-[^18^F]fluorobenzylazide.

## Data Availability

Not applicable.

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
