# Peer review of "Radiochemical Synthesis of 4-[18F]FluorobenzylAzide and Its Conjugation with EGFR-Specific Aptamers"

_molecules, 2022, doi:10.3390/molecules28010294_

Round 1

Reviewer 1 Report

The topic is intensively studying worldwide. The authors also published, as they also cited here, several research studies too. This manuscript well fits in that publishing trend. Application of aptamers has wide scale and welcome any new informations, discoveres by researchers. The authors pesented a little, but new step in this road. The 18F-labelled GR20 and GOL1 aptamers could be a potential biomarker for the PET diagnostic, hopefully. By the way I want to ask the authors to show the reason of their choose, why could be good or eminent candidate  the GR20 and GOL1 aptamers for the glioma imaging.

Author Response

Dear reviewer!

We are grateful to your helpful remarks. Hope our revised manuscript became better. Answering your question about aptamers used in research,- it's known that gliomas (different types of it) are highly represented with EGFR receptor and its mutant isoform, for example. Because of sighnificant aptamers affinity to EGFR (as we observed in literature and according to previous works of our co-authors), we consider they may be good for imaging.

Reviewer 2 Report

The manuscript by V.A. Il’in et al. submitted to Molecules described the synthesis of 18F conjugates of GR20 and GOL1 aptamers (18F-FB-GR20 and 18F-FB-GOL1). The interpretations and conclusions are justified by the results. The manuscript seems to be of interest for chemist and biologist working in the area. In general, with some major revisions, the manuscript is qualified to be published on Molecules.

Comments:

(1)   P2, line 68, “They utilize” should be “They utilized”.

(2)   P2, line 83, “radio tracer” should be “radiotracer”.

(3)   P6, line 185, “37°C” should be “37 °C”.

(4)   P9, line 284, “37°C” should be “37 °C”.

(5)   P9, line 290, “40°C” should be “40 °C”.

(6)   How about are the specific activities of 18F-FB-GR20 and 18F-FB-GOL1?

(7)   In particular, there are several format issues in References section. 

Author Response

Dear reviewer!

We are grateful to your helpful remarks. Hope our revised manuscript became better. 

According to your remarks:
P2, line 68, “They utilize” should be “They utilized” - we agree, now corrected

P2, line 83, “radio tracer” should be “radiotracer” - we agree, now corrected.

P6, line 185, “37°C” should be “37 °C” - we agree, now corrected.

P9, line 284, “37°C” should be “37 °C” - we agree, now corrected.

P9, line 290, “40°C” should be “40 °C” - we agree, now corrected.

How about are the specific activities of 18F-FB-GR20 and 18F-FB-GOL1? - we agree, now provided.

In particular, there are several format issues in References section. - we agree, now corrected.

Reviewer 3 Report

Review of “Radiochemical synthesis of 4-[18F]-fluorobenzyl azide and its conjugation with EGFR-specific aptamers”

Synopsis

Potential PET imaging agents of conjugates of GR20 and GOL1 were synthesized and tested. The short sequence target molecules (GR20 and GOL1) are known to be associated with the increase of epidermal growth factor receptors (EGFR), a known characteristic of glioblastoma. These novel radiotracers were synthesized via 4-[18F]-fluorobenzyl azide and then conjugated to the target moiety via a Cu(I)-catalyzed azide-alkyne cycloaddition. A fully automated synthesis for these radiotracers has been reported.

Major Corrections

·         Inconsistency with citations. References cited incorrectly (Ref#6). Please be uniform and consistent with format of references; some citations issues are as follows:

o   volume(issue):pg # (ref #3), while others us the form Vol.# No.# (ref #20).

o   et. al. is not consistently used throughout citations, used after one author (ref#1) or four authors (ref#18)

o   Form interchanges with: Last name, first initial and middle initial (Ref#1); first name-middle initial-last name (Ref#3); First initial. Last name (Ref#5)

o   No punctuation between titles and journals and inconsistency in capitalization (Ref#18)

Minor Corrections

  • Please refer/use the nomenclature as stated in the guideline https://doi.org/10.1016/j.nucmedbio.2017.09.004. (H.H. Coenen, et. al.; Nuclear Medicine and Biology, 2017, 55, v-xi.). As an example, please reframe from using ‘[18F]F-‘ and ‘18F’ (e.g. line 81) in this format; please follow this for all nomenclature discussed in aforementioned reference in your manuscript. Please superscript the isotope numeric value for all elements when the number value precedes the letter, e.g. [18F]; please use a dash without superscript when number follows the letter, e.g. F-18.

Abstract

·         Line 28 – Please use nomenclature cited above in minor corrections.

1. Introduction

·         Line 35 – Please supply a reference to the first statement.

·         Line 37 – Please remove phrase ‘At the same time’

·         Line 39 – Please reference the survival rate.

·         Line 42 – Please rephrase to ‘In the case of PET,’

·         Line 62 – Please refer to reference on nomenclature, shown above for up-to-date use of F-18

·         Line 69 – Please rephrase sentence ‘In [8] the authors synthesized a radioactive tracer, 4-[18F]-fluorobenzyl azide, for further conjugation with an aptamer.’

·         Line 83 – Please change ‘radio tracer’ to ‘radiotracer’

2. Results

·         Line 91 – Please state the amount of activity at start of synthesis.

·         Line 96 – Figure 1. Please state conditions for chromatogram

·         Line 114 – Please provide name before using the acronym of CuAAC

·         Line 121 – Figure 3. Please list conditions for chromatogram

·         Line 125 – Figure 4. Please list conditions for the chromatogram

·         Please provide a  molar activity and mass of F-19 compound produced from each labeling method. This value would also be important for the initial radiolabeling to provide 4-[18F]-fluorobenzyl azide

·         Line 135 – It is mentioned that the radiolabeled aptamer is “quite stable”; please provide how this conclusion was obtained, e.g. was stability achieved after 1-8 hours at room temperature via HPLC analysis?

3. Discussion

·         Line 144 – Please re-write introductory sentence

·          Line 174 through 177 – Suggestion to combine thought into one sentence

·          Line 207 – It is unclear to the Reviewer who “They” are referring to; please give more context to this description

4. Materials and Methods

·         Line 232 – The patent # is omitted

·         Line 259 – Please use reference above for correct nomenclature of radiotracer; e.g. [18F]fluorobenzyl azide

·         Figure 7 – Reviewer had difficulty reading captions from Figure 7.

·         Line 270 through 284 – Please reformat/edit the crucial write-up of the copper conjugation. Please state a range of what radioactivity was used in these experiments. Please provide how this was achieved through the automated module as shown in Figure 7.

5. Conclusion

·         Line 300 – Please clarify if this stability was only achieved with the use of TBTA additive

References

·         Line 325 – Please correct citation entry for journal and volume as: Zhurnal voprosy neirokhirurgii imeni N. N. Burdenko. 84(3), 113 - 118

·         Line 334 – Incorrect citation, when typing in refence into SciFindern as stated, lead to Caldeira, C.F., et. al. Circadian rhythms of hydraulic conductance and growth are enhanced by drought and improve plant performance. Nature Communications, 2014, 5, 5365; doi:10.1038/ncomms6365. Please update reference

·         Line 341 – Please correct document identification for reference 9 to ‘10.2967/jnumed.110.079418’

·         Line 359 – Please add pg# 2853-2855 and please fix document identification to 10.1021/ol0493094

·         Line 363 – Add pg# 438-449

·         Line 365 – Add pg# 1012-1024

Author Response

Dear reviewer!
We are grateul to your critical and valuable remarks! Hope our revised manuscript became better. 

Major Corrections:
Inconsistency with citations - we agree, now corrected.

Minor corrections:
Please refer/use the nomenclature as stated in the guideline https://doi.org/10.1016/j.nucmedbio.2017.09.004 - we agree, now corrected.

Abstract
Line 28 – Please use nomenclature cited above in minor corrections - we agree, now corrected.

  1. Introduction

Line 35 – Please supply a reference to the first statement - we agree, now corrected.

Line 37 – Please remove phrase ‘At the same time’ - we agree, now corrected.

Line 39 – Please reference the survival rate - we agree, now corrected.

Line 42 – Please rephrase to ‘In the case of PET’  - we agree, now corrected.

Line 62 – Please refer to reference on nomenclature, shown above for up-to-date use of F-18  - we agree, now corrected.

Line 69 – Please rephrase sentence ‘In [8] the authors synthesized a radioactive tracer, 4-[18F]-fluorobenzyl azide, for further conjugation with an aptamer.’  - we agree, now corrected.

Line 83 – Please change ‘radio tracer’ to ‘radiotracer’ - we agree, now corrected.

  1. Results

Line 91 – Please state the amount of activity at start of synthesis - we agree, now corrected.

Line 96 – Figure 1. Please state conditions for chromatogram - we agree, now corrected.

Line 114 – Please provide name before using the acronym of CuAAC -we agree, now provided.

Line 121 – Figure 3. Please list conditions for chromatogram - we agree, now corrected.

Line 125 – Figure 4. Please list conditions for the chromatogram - we agree, now corrected.

Please provide a  molar activity and mass of F-19 compound produced from each labeling method. This value would also be important for the initial radiolabeling to provide 4-[18F]-fluorobenzyl azide - we agree, now provided;
      Line 135 – It is mentioned that the radiolabeled aptamer is “quite stable”; please provide how this conclusion was obtained, e.g. was stability achieved after 1-8 hours at room temperature via HPLC analysis? - we agree, now corrected.

  1. Discussion

Line 144 – Please re-write introductory sentence - we agree, now corrected.

Line 174 through 177 – Suggestion to combine thought into one sentence - we agree, now corrected.

 Line 207 – It is unclear to the Reviewer who “They” are referring to; please give more context to this description - we agree, now corrected.

  1. Materials and Methods

Line 232 – The patent # is omitted - we agree, now corrected.

Line 259 – Please use reference above for correct nomenclature of radiotracer; e.g. [18F]fluorobenzyl azide - we agree, now corrected.

 Figure 7 – Reviewer had difficulty reading captions from Figure 7 - we agree, now corrected.
Line 270 through 284 – Please reformat/edit the crucial write-up of the copper conjugation. Please state a range of what radioactivity was used in these experiments. Please provide how this was achieved through the automated module as shown in Figure 7 - we agree, now provided and corrected.

  1. Conclusion

Line 300 – Please clarify if this stability was only achieved with the use of TBTA additive-we agree, now corrected.

References

Line 325 – Please correct citation entry for journal and volume as: Zhurnal voprosy neirokhirurgii imeni N. N. Burdenko. 84(3), 113 - 118 - we agree, now corrected.

Line 334 – Incorrect citation, when typing in refence into SciFindernas stated, lead to Caldeira, C.F., et. al. Circadian rhythms of hydraulic conductance and growth are enhanced by drought and improve plant performance. Nature Communications20145, 5365; doi:10.1038/ncomms6365. Please update reference - we agree, now updated.

Line 341 – Please correct document identification for reference 9 to ‘10.2967/jnumed.110.079418’ - we agree, now corrected.

Line 359 – Please add pg# 2853-2855 and please fix document identification to 10.1021/ol0493094 - we agree, now added.

Line 363 – Add pg# 438-449 -we agree, now added.

Line 365 – Add pg# 1012-1024 -we agree, now added.

Round 2

Reviewer 3 Report

Reviewer feels the authors have done an excellent job adjusting manuscript from all prior suggestions. It is recommended to Accept as is.